# A Mendelian randomization study of the effects of blood lipids on breast cancer risk

Christoph Nowak [1] & Johan Ärnlöv[1,2]

Observational studies have reported inconsistent associations between circulating lipids and breast cancer risk. Using results from >400,000 participants in two-sample Mendelian randomization, we show that genetically raised LDL-cholesterol is associated with higher risk of breast cancer (odds ratio, OR, per standard deviation, 1.09, 95% confidence interval, 1.02–1.18, $P = 0.020$) and estrogen receptor (ER)-positive breast cancer (OR 1.14 [1.05–1.24] $P = 0.004$). Genetically raised HDL-cholesterol is associated with higher risk of ER-positive breast cancer (OR 1.13 [1.01–1.26] $P = 0.037$). HDL-cholesterol-raising variants in the gene encoding the target of CETP inhibitors are associated with higher risk of breast cancer (OR 1.07 [1.03–1.11] $P = 0.001$) and ER-positive breast cancer (OR 1.08 [1.03–1.13] $P = 0.001$). LDL-cholesterol-lowering variants mimicking PCSK9 inhibitors are associated ($P = 0.014$) with lower breast cancer risk. We find no effects related to the statin and ezetimibe target genes. The possible risk-promoting effects of raised LDL-cholesterol and CETP-mediated raised HDL-cholesterol have implications for breast cancer prevention and clinical trials.

[1] Division of Family Medicine and Primary Care, Department of Neurobiology, Care Sciences and Society (NVS), Karolinska Institutet, Alfred Nobels Allé 23, SE-14152 Huddinge, Sweden. [2] School of Health and Social Studies, Dalarna University (Högskola Dalarna), SE-79188 Falun, Sweden. Correspondence and requests for materials should be addressed to C.N. (email: christoph.nowak@ki.se)

**B**reast cancer affects up to 1 in 8 women during their lifetime and is the second leading cause of death among women in the Western world[1,2]. Cardiovascular diseases and breast cancer share many metabolic risk factors, including diet, obesity and physical activity[3]. The role of lipids, particularly in estrogen receptor-positive (ER-positive) breast cancer, is well known, but causal pathways have been difficult to disentangle[4]. Observational studies on associations between blood lipids and breast cancer have yielded equivocal results, with suggestive associations between raised triglyceride and high-density lipoprotein-cholesterol (HDL-cholesterol) and lower risk of breast cancer[5,6], that have, however, not been confirmed by other studies[7] and may depend on menopausal status[8]. No evidence for an association between low-density lipoprotein-cholesterol (LDL-cholesterol)[5–7,9] or use of statins (widely prescribed LDL-cholesterol-lowering drugs) and breast cancer has been detected[10–12], although in women with breast cancer, statins may be associated with lower recurrence risk[13] and reduced breast cancer-specific mortality[14].

The effects of other lipid-modifying drugs on breast cancer are less well studied. No association with cancer occurrence was reported in the largest trial of an HDL-cholesterol-increasing cholesteryl ester transfer protein (CETP) inhibitor[15]. Kobberø Lauridsen et al.[16] found no association between genetic variants in the NPC1L1 gene (encoding the target of ezetimibe) and cancer in ~67,000 persons. No concerns have been raised about associations between proprotein convertase subtilisin/kexin type 9 (PCSK9) inhibitors and cancer risk[17].

Limited follow-up time in clinical trials and confounding or reverse causation in observational studies render conclusions about causality uncertain. In epidemiologic settings, Mendelian randomization (MR) has been developed to assess causality[18]. Parental genetic variants are randomly inherited, and MR uses variants that are associated with an exposure as instruments to test for associations with an outcome. This concept is analogous to randomized designs and minimizes bias from confounding and reverse causation. It can also predict drug effects by using mutations in drug target genes as instruments[19]. MR makes several assumptions that are often difficult to ascertain, including the absence of genetic effects on the outcome that are independent of the exposure (absent horizontal pleiotropy)[20]. It can thus only provide preliminary evidence of causality that may inform subsequent intervention studies, drug monitoring and public health approaches[21].

MR studies have demonstrated, for instance, an inverse association between genetically predicted obesity and risk of breast cancer[22,23]. Orho-Melander et al.[24] studied ~16,000 women and found suggestive effects of raised HDL-cholesterol and reduced triglycerides on increased breast cancer risk that did not, however, reach nominal significance. In the same study, the LDL-lowering allele of a variant in the statin target gene HMGCR was associated with lower risk of breast cancer while an LDL-cholesterol genetic score was not, suggesting an LDL-independent mechanism. To our knowledge, MR has not been applied to study effects of lipids on breast cancer risk in the largest available genetic datasets from the Global Lipid Genetics Consortium (GLGC)[25] and the 2017 release of the Breast Cancer Association Consortium (BCAC)[26] of over 180,000 participants each.

In this study, we use two-sample MR in the largest available genetic datasets to assess causal associations between circulating LDL-cholesterol, HDL-cholesterol, triglycerides and variants in genes encoding lipid-modifying drug targets on the risk of total breast cancer, ER-positive and ER-negative breast cancer. We find possible risk-increasing effects of raised LDL-cholesterol and CETP-mediated raised HDL-cholesterol that may have implications for breast cancer prevention.

## Results

**Study overview.** Figure 1 summarizes the study flow. Individual genetic variant associations are listed in Supplementary Tables 1 and 2. Full MR results are available in Supplementary Tables 3–6 for lipids and in Supplementary Table 7 for drug targets. Causal estimates are expressed as odds ratios (OR) and 95% confidence interval (CI) per standard deviation increment in plasma lipid level. Comprehensive MR refers to using all variants associated with the target lipid, whilst restrictive MR excludes variants associated with any of the other lipids ($P < 0.001$).

**Effect of lipid levels.** In comprehensive MR (Supplementary Table 3), we detected suggestive associations between raised LDL-cholesterol and breast cancer risk ($P = 0.055$). Genetically raised HDL-cholesterol was associated with breast cancer risk ($P = 0.003$) and ER-positive disease risk ($P = 0.002$). Raised triglycerides were negatively associated with all three outcomes. There was evidence of heterogeneity in all analyses ($Q'$ $P$-values $< 10^{-5}$). Following exclusion of pleiotropic variants in restrictive MR (Supplementary Table 4), raised LDL-cholesterol was associated with higher risk of any breast cancer (OR 1.12, 95% CI, 1.02–1.23, $P = 0.017$) and ER-positive breast cancer (OR 1.17, 95% CI, 1.05–1.29, $P = 0.004$) with consistent estimates across the Egger and median methods but evidence of remaining heterogeneity ($Q'$ $P$-values $< 10^{-4}$). Raised HDL-cholesterol had no clear association with breast cancer risk (OR 1.08, 95% CI, 0.96–1.21, $P = 0.198$) with significant remaining heterogeneity ($Q'$ $P$-value = 0.003). There was evidence of an effect of raised HDL-cholesterol on increased ER-positive breast cancer risk (OR 1.13, 95% CI, 1.01–1.26, $P = 0.028$, $Q'$ $P$-value = 0.169). Triglycerides were not associated with any of the outcomes ($P > 0.4$).

We applied the MR-PRESSO method to the restrictive MR models to identify and remove outlier variants followed by retesting for heterogeneity (Fig. 2, Supplementary Table 5). In inverse variance-weighted MR following the removal of outliers, raised LDL-cholesterol had a risk-increasing effect on breast cancer (OR 1.09, 95% CI, 1.02–1.18, $P = 0.020$, $Q'$ $P$-value = 0.102) and ER-positive breast cancer (OR 1.14, 95% CI, 1.05–1.24, $P = 0.004$, $Q'$ $P$-value = 0.124) and no association with ER-negative disease ($P = 0.577$). Raised HDL-cholesterol had no nominally significant association with either breast cancer risk (OR 1.07, 95% CI, 0.97–1.19, $P = 0.171$, $Q'$ $P$-value = 0.090) or ER-negative disease (OR 1.09, 95% CI, 0.91–1.30, $P = 0.365$, $Q'$ $P$-value = 0.108), but appeared associated with increased risk of ER-positive disease (OR 1.13, 95% CI, 1.01–1.26, $P = 0.037$, $Q'$ $P$-value = 0.169). Genetically predicted triglyceride levels were not associated with any of the outcomes. Application of MR-PRESSO to the comprehensive selection without exclusion of variants associated with other lipids produced causal estimates in the same direction, but there was significant ($P < 0.05$) remaining heterogeneity after exclusion of outliers in all cases (Supplementary Table 6).

**Effect of lipid-modifying drug targets.** We implemented inverse variance-weighted MR and MR Egger with consideration of the correlation between genetic variants using seven LDL-cholesterol-associated variants in PCSK9, three LDL-associated variants each for NPC1L1 and HMGCR, and six variants for LDLR (Supplementary Table 2). For CETP, we selected 11 HDL-cholesterol-associated variants. LDL-raising variants in PCSK9 were associated with increased risk of breast cancer (OR 1.10, 95% CI, 1.02–1.19, $P = 0.014$) but not with ER-positive (OR 1.08, 95% CI, 0.99–1.18, $P = 0.099$) or ER-negative breast cancer (OR 1.13, 95% CI, 0.98–1.30, $P = 0.089$) at the nominal significance level (Fig. 3, Supplementary

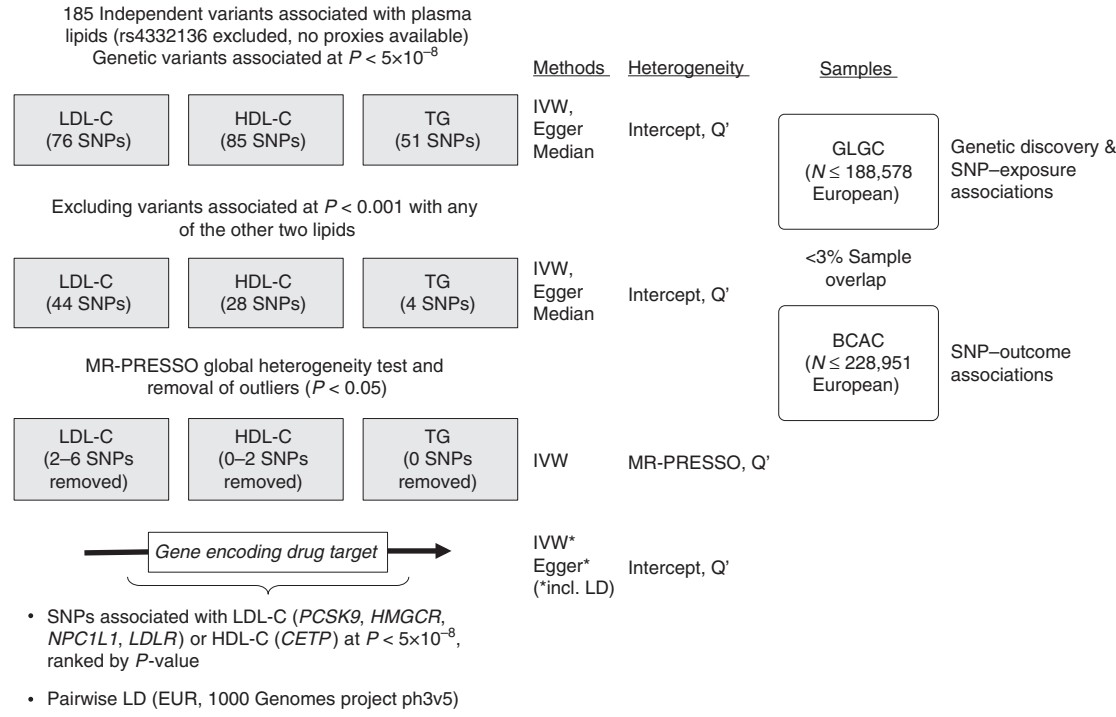

**Fig. 1** Study overview. BCAC: Breast Cancer Association Consortium, Egger MR Egger method, EUR European reference sample, GLGC Global Lipids Genetics Consortium, HDL-C high-density lipoprotein-cholesterol, Intercept Egger regression intercept term, IVW inverse variance-weighted method, LD linkage disequilibrium, LDL-C low-density lipoprotein-cholesterol, MR-PRESSO MR pleiotropy residual sum and outlier, Q' modified 2nd order weight heterogeneity test, SNP single nucleotide polymorphisms, TG triglycerides

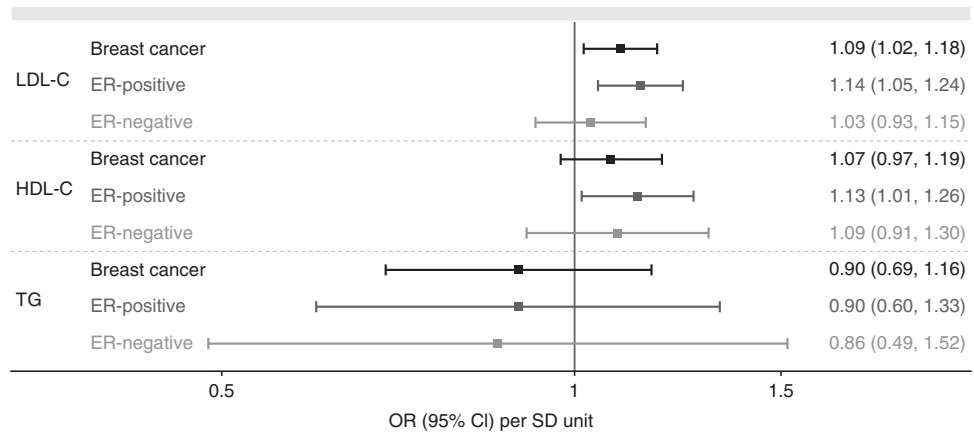

**Fig. 2** Causal estimates of blood lipid levels on risk of all, ER-positive and ER-negative breast cancer. Inverse variance-weighted instrumental variable analysis using genome-wide significantly associated independent variants as instrumental variables for each lipid. Results following exclusion of variants associated at $P < 0.001$ with any of the other lipids and following removal of outlier variants ($P < 0.05$ in MR-PRESSO) are shown. Causal estimates express the change in odds ratio (OR) per standard deviation (SD) increment in lipid concentration. Error bars indicate 95% confidence intervals

Table 7). HDL-raising variants in *CETP* were associated with raised breast cancer risk (OR 1.07, 95% CI, 1.03–1.11, $P = 0.001$) and ER-positive breast cancer risk (OR 1.08, 95% CI, 1.03–1.13, $P = 0.001$), although in both cases, MR Egger was suggestive of a null association (OR 1.00, 95% CI, 0.87–1.15, $P = 0.972$; and OR 1.01, 95% CI, 0.86–1.17, $P = 0.948$, respectively). The associations with ER-negative breast cancer risk did not reach nominal significance (OR 1.07, 95% CI, 0.99–1.15, $P = 0.075$). Estimates for *NPC1L1* were suggestive of risk-increasing effects of raised LDL-cholesterol on breast cancer

and ER-positive breast cancer risk in inverse variance-weighted analysis, but MR Egger estimates were in the opposite direction, casting doubt on the validity of MR assumptions in this case. LDL-cholesterol-raising variants in *HGMCR* had a suggestive association with breast cancer risk (OR 1.16, 95% CI, 0.98–1.37, $P = 0.086$; Fig. 3). The analysis for LDLR variants provided inconsistent estimates across methods and implied unaccounted-for pleiotropy in the Egger intercept test for all three outcomes, cautioning against an interpretation of the observed results in inverse variance-weighted MR.

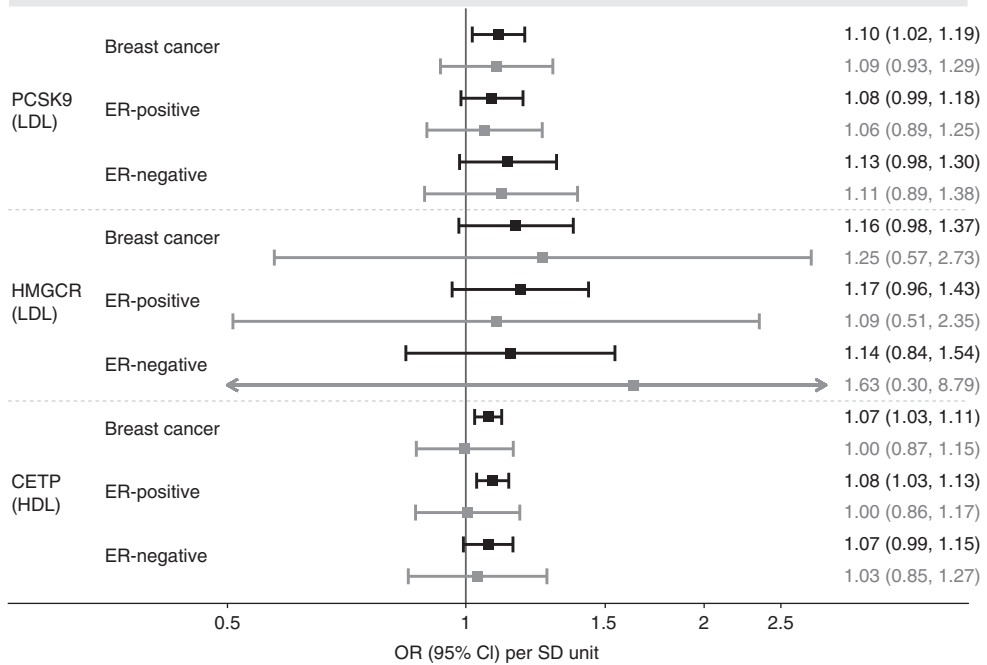

**Fig. 3** Causal estimates of blood lipid level-increase due to genetic variants in genes encoding drug targets. Inverse variance-weighted (black) and MR Egger (gray) instrumental variable estimates using genome-wide significantly associated variants within 100 b either side of the gene in low linkage disequilibrium ($r^2 < 0.4$). Analyses take correlations between genetic instruments into account. Causal estimates express the change in odds ratio (OR) per standard deviation (SD) increment in lipid concentration. Error bars indicate 95% confidence intervals

## Discussion

In two-sample summary-level MR, we found an association between genetically raised LDL-cholesterol and increased risk of breast cancer and ER-positive breast cancer. Instrumental variable analysis with HDL-cholesterol-raising variants in *CETP* suggested a small risk-increasing effect on breast cancer and ER-positive disease; however, possible bias from pleiotropy cannot be excluded. Lowered LDL-cholesterol due to variants in *PCSK9* had a suggestive protective effect on breast cancer risk.

The risk-increasing effect of genetically raised LDL-cholesterol on total and ER-positive breast cancer contrasts with observational studies that reported inconsistent results generally suggesting a null association, with marked heterogeneity depending on study design, menopausal status and body mass index of participants[5–7,9]. In our study, we were able to minimize pleiotropic effects that bias observational studies and found evidence of an LDL-specific harmful effect on ER-positive and (to a lesser extent) total breast cancer risk. The inverse association between triglycerides and breast cancer risk in comprehensive MR was abolished after excluding variants associated with other lipids, implying that triglyceride levels do not affect breast cancer risk independently. This result broadly concurs with observational studies that reported either absent or weak inverse associations[6,7,27].

Genetically raised HDL-cholesterol was associated with increased risk of ER-positive breast cancer. Observational studies have reported inconsistent results on associations between HDL and breast cancer, including null effects[5,7,27,28], inverse associations in post-menopausal women[6] and unidirectional associations in pre-menopausal women[29]. The effect of HDL-cholesterol on breast cancer risk in our study concurs with a non-significant association in an earlier genetic study[24], and the absence of a stronger effect in our summary-level data may relate to lack of power and an inability to stratify participants by menopausal status. Another explanation could be different effects depending on the metabolic health of the individual, as in vivo studies found evidence that oxidation status of HDL-cholesterol in a normo- or hyperlipidaemic context may determine its effects on the promotion of breast cancer metastasis[30].

Taken together, laboratory studies have demonstrated that lipoprotein fractions affect breast cancer growth both directly and as precursors for cholesterol metabolites[30,31], and observational studies in women have hitherto not been able to consistently define potential effects. Our study provides genetic evidence of a harmful association between raised LDL-cholesterol and breast cancer occurrence, as well as a suggestive harmful effect of raised HDL-cholesterol. A deficit of our study is the inability to stratify women by menopausal status. The endocrine changes of the menopause likely affect plasma lipid composition and the interaction with breast tissue. For instance, a meta-analysis of observational studies found that an inverse association between HDL-cholesterol and breast cancer was only present in post- but not in premenopausal women[6]. Differing effects depending on menopausal status are further suggested by findings that genetically predicted obesity is inversely associated breast cancer risk, which contrasts with a positive association between obesity and postmenopausal breast cancer risk in observational studies[22]. Large biobanks such as the UK Biobank may in the future allow to further dissect the suggestive causal effects of HDL-cholesterol on breast cancer discovered in our study.

Raised HDL-cholesterol due to genetic variants in *CETP* was associated with raised total and ER-positive breast cancer risk, but only two of the 11 variants were individually associated with breast cancer at the nominal significance level and MR Egger implied absent effects (Supplementary Tables 2 and 6). Whether pharmacological CETP inhibition could affect breast cancer risk remains uncertain, as lifelong genetic effects and consequences of pharmacological intervention in mid-life may differ[21]. None of the four clinical trials of CETP inhibitors published by May 2018 have reported associations with breast cancer occurrence, although the proportion of women (19.2% across trials) and the short follow-up of up to 4.1 years pose limitations on the

| Table 1 Breast cancer outcomes in clinical trials of CETP inhibitors | | |
|---|---|---|
| **Study name** | **Description** | **Reported outcomes related to breast cancer** |
| ILLUMINATE[47] | Torcetrapib, $N = 15,067$ high cardiovascular risk, 1–2-year follow-up, 1,679 women in active and 1,673 women in comparator group | No specific reporting on breast cancer. There was 1 cancer death in the torcetrapib group and 0 in the comparator group. Serious adverse events affecting the "reproductive system or breast" occurred in 27 active and 18 control persons. |
| dal-OUTCOMES[48] | Dalcetrapib, $N = 15,871$ post-acute coronary syndrome, 31-month follow-up, 1,573 women in active and 1,497 women in comparator group | No specific reporting on breast cancer. Malignant or unspecific tumours occurred in 270 persons (48 fatal) in the active group and 286 persons (47 deaths) in the comparator group. |
| ACCELERATE[49] | Evacetrapib, $N = 12,092$ high vascular risk, 26-month follow-up, 1,390 women in active and 1,394 women in comparator group | No reporting of breast cancer. |
| REVEAL[15] | Anacetrapib, $N = 30,449$ high vascular risk, 4.1-year follow-up, 2,459 women in active and 2,456 women in comparator group | Breast cancer occurred in 24 persons in the active, and 27 persons in the comparator group. |

detection of potential cancer-related effects. Table 1 summarizes these trials.

Lowering of LDL-cholesterol due to variants in *PCSK9* was associated with risk-reducing effects on breast cancer occurrence in our study. A 2017 review of clinical studies comparing PCSK9 inhibitors to placebo found no association with risk of any cancer, although the direction of association (OR 0.91, 95% CI, 0.63–1.31) does not exclude a possibly protective effect[19]. A similar non-significant protective association with cancer risk was found in a phenome-wide association study of genetic variants in *PCSK9* in the UK Biobank sample[32].

The potential, but not nominally significant effect ($P = 0.086$) of variants in the gene encoding the target of statins could point to a true signal that our study was underpowered to detect or that may differ in subgroups (such as postmenopausal women) that we were not able to assess. The possible risk-reducing effect of statin mimicry in MR chimes with observational studies that reported either null or protective associations with breast cancer risk and mortality[10–14]. Future studies with genetic and drug exposure data that allow analysis in subgroups should address any possible effects of statins on breast cancer.

An early clinical trial of ezetimibe raised concerns that combination therapy with statins might be associated with increased risk of cancer. Subsequent longer follow-up and comparisons across other clinical trials, however, found no association with raised cancer risk[33,34]. Our findings indicate a possible protective effect and agree with an earlier smaller genetic study[16], but inconsistent estimates between Egger and inverse variance-weighted MR in our analysis imply violations of model assumptions that do not allow for conclusive interpretation.

Strengths of our study include the use of the largest available summary genetic datasets and extensive diagnostics to evaluate the validity of MR assumptions and limit the potential for bias from pleiotropy. Limitations include our inability to replicate results in independent datasets, concerns about pleiotropy from (un)measured confounders, possible weak instrument bias and lack of power for drug analyses. A bias toward the null because of Winner's curse[35], as genetic discovery had been implemented in the same dataset used to estimate exposure associations, cannot be excluded. MR assesses the life-long effects of genetic variation and cannot be directly compared to pharmacological inhibition in adult life. The analyses accounted for population stratification (genetic principle components and restriction to European ethnicity) and pleiotropy (MR Egger), but remaining sources of bias such as canalization cannot be ruled out. Finally, we could not assess the influence of menopausal status and our results only apply to women of European ethnicity.

## Methods

**Summary genetic association data**. Genome-wide association study results in persons of mostly European ancestry were obtained from the GLGC (up to 188,577 persons) for plasma lipids[25] and from BCAC for risk of breast cancer (up to 122,977 affected and 105,974 control women)[26]. Both studies included rigorous quality control, imputation to the 1000 Genomes Project panel and adjustments for age and population structure. The studies have existing ethical permissions from their respective institutional review boards and include participant informed consent. The outcomes in the present study were risk of any breast cancer, ER-positive and ER-negative breast cancer as defined in BCAC[26].

**Analyses in the Global Lipids Genetics Consortium**. Persons of European ancestry from 47 studies genotyped with different genome-wide association study arrays ($n = 94,595$) or on the Metabochip array ($n = 93,982$) with imputation to the 1000 Genomes Project reference were studied. In most included studies, blood lipid concentrations had been measured after an >8 h fast. Participants on lipid-lowering medications were excluded. Traits were adjusted for age, age-squared, sex and principle components, as well as quantile-normalized within each cohort. For genetic association analysis by linear regression, lipid levels were inverse normal-transformed and cohort-wise results combined in fixed effect meta-analysis[25].

**Analyses in the Breast Cancer Association Consortium**. The consortium genotyped on the OncoArray altogether 61,282 women with breast cancer and 45,494 control women without breast cancer of European ancestry who were enrolled in 68 studies in the BCAC and the Discovery, Biology and Risk of Inherited Variants in Breast Cancer Consortium (DRIVE). Genotypes were imputed to ~21 million variants using the 1000 Genomes Project (Phase 3) reference panel. Variants with minor allele frequency <0.5% and imputation quality score <0.3 were excluded resulting in ~11.8 million variants for logistic regression analysis adjusted for genetic principle components and country. Results were combined in fixed-effect meta-analysis with results from the Collaborative Oncological Gene-environment Study (iCOGS, 46,785 cases and 42,892 controls) and 11 other breast cancer genome-wide association studies (14,910 cases and 17,588 controls). The current study uses summary results from women of European ancestry[26].

**Genetic instruments for blood lipids**. We extracted association statistics for LDL-cholesterol, HDL-cholesterol and triglycerides in GLGC for 185 genetic variants in 157 loci previously demonstrated to be associated with at least one lipid fraction[36]. We constructed two genetic instruments for each lipid. First, we selected all genome-wide significant ($P < 5 \times 10^{-8}$) variants associated with each lipid for comprehensive MR (76 variants for LDL-cholesterol, 85 for HDL-cholesterol, 51 for triglycerides). Second, to reduce possible pleiotropic effects we excluded in each selection those variants that were associated at $P < 0.001$ with any of the other two lipids for restrictive MR (44 variants for LDL-cholesterol, 28 for HDL-cholesterol, 4 for triglycerides).

**Proxies for drug targets**. To assess potential causal effects of changes in lipid levels due to pharmacological intervention, we selected polymorphisms within ±100 base pairs of genes encoding drug targets that were genome-wide significantly associated with the target lipid and in low linkage disequilibrium with each other. Variants were ranked by $P$-value for lipid association in GLGC and iteratively selected in the order of increasing $P$-value provided they were in low linkage disequilibrium ($r^2 < 0.4$) with variants selected in preceding steps. We obtained pairwise linkage disequilibrium based on Phase 3 (Version 5) of the 1000 Genomes Project combined European reference sample via LDlink[37]. We used associations with HDL-cholesterol to construct the genetic instrument for *CETP* and associations with LDL-cholesterol to construct instruments for *HMGCR* (encodes the

target of statins), *PCSK9*, *NPC1L1* (encodes the target of ezetimibe) and *LDLR*. *LDLR* does not mimic a drug target but is a common site of mutations causing familial hypercholesterolemia (OMIM 606945) and was included to assess the role of the LDL receptor pathway.

**Mendelian randomization**. One variant (rs4332136) was not available and excluded as no proxy variant ($r^2 > 0.8$) was available. Genetic effects were aligned to the lipid-increasing allele and alignment checked by comparing the minor allele frequencies reported by BCAC and GLGC. Palindromic variants that could not be unambiguously aligned and multi-allelic variants with different effect and reference alleles in BCAC and GLCL were removed.

We used inverse variance-weighted, Egger and weighted median MR to assess causal effects of lipid fractions. The inverse variance-weighted method regresses genetic associations with the outcome on associations with lipid levels and fixes the intercept at zero. In the absence of directional pleiotropy, it provides robust causal estimates[38]. MR Egger allows free estimation of the intercept, although further assumptions, such as the independence between instrument strength and direct effects, cannot be easily verified. A statistically significant intercept term implies the presence of unbalanced pleiotropy and causal estimates in MR Egger are less precise than those in inverse variance-weighted MR[39]. Weighted median MR allows some variants to be invalid instruments provided at least half are valid instruments. It uses inverse variance weights and bootstrapping to estimate CIs[40]. Figure 1 provides an overview of the methods used in this study. For drug target MR with variants in moderate linkage disequilibrium ($r^2 < 0.4$), we implemented the inverse variance-weighted and Egger methods with explicit modelling of correlations between genetic variants according to the method suggested Burgess et al.[41] as implemented in the MendelianRandomization software in R[42]. To assess heterogeneity between individual genetic variants' estimates, we used the Egger intercept test[43], the $Q'$ heterogeneity statistic[44] and the MR pleiotropy residual sum and outlier (MR-PRESSO)[45] test. The $Q'$ statistic uses modified 2nd order weights that are a derivation of a Taylor series expansion and take into account uncertainty in both numerator and denominator of the instrumental variable ratio (this eases the no-measurement-error, NOME, assumption)[44]. The MR-PRESSO framework relies on the regression of variant-outcome associations on variant-exposure associations and implements a global heterogeneity test by comparing the observed distance (residual sums of squares) of all variants to the regression line with the distance expected under the null hypothesis of no pleiotropy[45]. In case of evidence of horizontal pleiotropy, the test compares individual variants' expected and observed distributions to identify outlier variants. We used an implementation of MR-PRESSO in R (https://github.com/rondolab/MR-PRESSO) with default parameters to (i) test for global heterogeneity; (ii) if significant at $P < 0.05$ identify and remove outliers; and (iii) retest to evaluate if outlier removal had resolved heterogeneity. We consider as results causal estimates that agree in direction and magnitude across all MR methods, pass nominal significance in inverse variance-weighted MR, and do not show evidence of bias from horizontal pleiotropy in heterogeneity tests. Analyses were carried out with the MendelianRandomization[42], TwoSampleMR and MR-PRESSO[45] packages in R version 3.3.2 (2016-10-31).

**Power**. We used mRnd (http://cnsgenomics.com/shiny/mRnd/) for post-hoc power calculations. At an alpha level of 0.05, we estimated 80% power to detect causal effects on breast cancer risk per standard deviation increment in lipid level of OR 1.06 (LDL-cholesterol), OR 1.07 (HDL-cholesterol) and OR 1.07 (trigly-cerides). The corresponding estimates for ER-positive breast cancer were OR 1.06 (LDL-cholesterol), OR 1.07 (HDL-cholesterol) and OR 1.08 (triglycerides); and the estimates for ER-negative breast cancer were OR 1.10 (LDL-cholesterol), OR 1.11 (HDL-cholesterol) and OR 1.12 (triglycerides), respectively.

**Sample overlap**. Participant overlap between the samples used to estimate genetic associations with the exposure and the outcome, respectively, in two-sample MR can bias results[46]. A careful comparison of the samples included BCAC and GLGC showed one common cohort (EPIC), which accounted for 2.9% of cases and 3.3% of control persons in BCAC, and for 1.7% (1.0% if only considering women) of participants in GLGC. Based on a simulation study of the association between sample overlap and the degree of bias in instrumental variable analysis[46], this degree of overlap (<5%) is unlikely to influence results in a meaningful way.

**Code availability**. The analysis code in R is available on request and all data displayed in figures are available in Supplementary Tables 1–7.

## Data availability
All summary genetic association data used in this study are available online, GLGC (http://lipidgenetics.org/) and BCAC (http://bcac.ccge.medschl.cam.ac.uk/).

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

## Acknowledgements

The breast cancer genome-wide association analyses were supported by the Government of Canada through Genome Canada and the Canadian Institutes of Health Research, the 'Ministère de l'Économie, de la Science et de l'Innovation du Québec' through Genome Québec and Grant PSR-SIIRI-701, The National Institutes of Health (U19 CA148065, X01HG007492), Cancer Research UK (C1287/A10118, C1287/A16563, C1287/A10710) and The European Union (HEALTH-F2-2009-223175 and H2020 633784 and 634935). All studies and funders are listed in Michailidou et al.[26]. Key software packages and analysis code were sourced from https://cran.r-project.org/web/packages/MendelianRandomization/index.html; https://github.com/MRCIEU/TwoSampleMR; and https://github.com/rondolab/MR-PRESSO. J.Ä. was supported by a grant from the Swedish Research Council (2012-2215). C.N. was supported by EFSD/Lilly (European Foundation for the Study of Diabetes Young Investigator Programme).

## Author contributions

C.N. conceived of and designed the study. C.N. analysed the data, wrote the first draft of the manuscript and is the guarantor of the study. J.Ä. contributed funding and critically revised the manuscript.

## Additional information

**Competing interests:** The authors declare no competing interests.

