## [Peer Review File · Nature Communications]

Reviewers' Comments:

Reviewer #1:

Remarks to the Author:

This is a very interesting article looking into association of lipid levels, metabolism with incidence of breast cancer. This is a very interesting and potentially clinically important observation

The authors use extensive analysis using mendelian randomization to explore associations between breast cancer and various genetic variants that have been associated with changes in lipid levels or response to specific drugs.

Authors appropriately acknowledge the limitations of their analysis and inability to test their findings on an independent data set.

The main limitation of this study is the choice of SNPs based on 1 study, upon literature review this approach has been shown to increase possibility of an error: Amy E. Taylor et al. Mendelian randomization in health research: Using appropriate genetic variants and avoiding biased estimates. I would welcome a more expert statistical review to answer if the authors chose the most appropriate tools to decrease potential for bias.

The authors rightly acknowledge the ongoing controversy about the role of HDL and lipids levels in general to the risk of developing breast cancer. A stronger point in discussion explaining discrepancies and how this research might push the field in the right direction would strengthen the article.

Minor:

please use estrogen or oestrogen consistently throughout the paper.

Reviewer #2:

Remarks to the Author:

Nowak et al. perform two sample summary-based multi-instrument MR on blood lipids and breast cancer risk using the largest GWAS summary statistic datasets to date for lipids (GLGC) and breast cancer. They show that genetically-raised HDL-C was associated with higher risk of breast cancer and estrogen receptor-positive breast cancer, with a specific genic effect on CETP variants. Genetically lowering LDL cholesterol in statin target genes, PCSK9 inhibitors and ezetimide may have protective effects.

The study is a well executed study that leverages the largest GWAS summary datasets to date for these traits. This work is relevant for breast cancer and has important implications for ongoing clinical trials. Below are my comments:

Strengths

1. leverages appropriate MR methods and approaches to infer causality of blood lipids on breast cancer
2. leverages largest GWAS summary datasets for both these traits
3. Well-written and concise

Limitations

4. the strength of statistical evidence of the causal relationship of HDL-C and breast cancer, and that of LDL-C and breast cancer, is consistent but the statistical evidence is moderate. One would have liked to see stronger evidence in the MR tests to more confidently say whether such causal relationships truly exist (re. more replication but its understood that this is sometimes difficult to find more datasets).

5. the restrictive MR approach excludes pleiotropy with other lipid traits using genome-wide significant cutoff. One could use a more relaxed P-value such as $P < 0.05$ to more strictly exclude pleiotropic effects on these traits.

6. proxies for drug targets - rather than using variants with LD $r^2 < 0.40$ to address correlated IVs, one can consider using explicit approaches that takes into account the correlation between IVs (see PMID: 26661904)

7. Cochran's Q test was used to test for heterogeneity between IVs. The original method has been shown to result in type 1 error rate (PMID: 29686387 and biorxiv: <https://www.biorxiv.org/content/early/2018/02/27/159442>). Please ensure that the modified version of the Q test (Q' test) is being used, which corrects for this inflated error rate.

8. Cooks distance was used to identify outliers; however, a recent study (PMID: 29686387) showed that Cooks distance performs less than ideal in detecting correct outliers; instead, one can use the Q' outlier test (<https://www.biorxiv.org/content/early/2018/02/27/159442>) or MR-PRESSO (PMID: 29686387)

9. After removal of outliers, one can again perform test of heterogeneity between IVs to ensure that heterogeneity is no longer significant after removal of outliers in the outlier-corrected MR test. This can provide reassurance that heterogeneity is not an issue when assessing the causal estimate in the outlier-corrected MR test.

10. Can the authors please confirm if there is (and if so, approximately how much) participant overlap between the GLGC dataset and the Breast cancer dataset. Overlap of samples can potentially increase the false positive rate in two-sample MR (see PMID: 27625185)

Minor comment

11. the reference for selecting 185 genetic variants in 157 loci should include PMID: 24097064

Reviewer #1:

This is a very interesting article looking into association of lipid levels, metabolism with incidence of breast cancer. This is a very interesting and potentially clinically important observation.

The authors use extensive analysis using mendelian randomization to explore associations between breast cancer and various genetic variants that have been associated with changes in lipid levels or response to specific drugs. Authors appropriately acknowledge the limitations of their analysis and inability to test their findings on an independent data set.

The main limitation of this study is the choice of SNPs based on 1 study, upon literature review this approach has been shown to increase possibility of an error: Amy E. Taylor et al. Mendelian randomization in health research: Using appropriate genetic variants and avoiding biased estimates. I would welcome a more expert statistical review to answer if the authors chose the most appropriate tools to decrease potential for bias.

Author response:

We thank the Reviewer for their constructive criticism and for raising important points about the design of our study. The Reviewer questions our selection of "SNPs based on 1 study" and cites Amy E. Taylor et al.'s study (*Econ Hum Biol.* 2014. PMID 24388127). We fully agree that selecting genetic instruments in 1 study is inappropriate and are grateful for the opportunity to better explain our selection. We selected SNPs in the Global Lipids Genetics Consortium dataset, which is a meta-analysis of over thirty separate studies/samples in mostly European participants. In the study by Taylor et al. (PMID 24388127) referred to by the Reviewer, the authors rightly criticize previous MR studies (specifically in the tobacco field) because "[...] using genetic variants selected based on their association in a **single sample** are likely to be misleading [...]" (bold emphasis added). The authors refer to a published MR study where SNPs had been selected in one sample and chance correlations between SNPs and confounders or population stratification may have biased SNP selection. Their criticism does not refer to combining **many independent samples** in a consortium like the GLGC, but rather - as we understand it - to selecting SNPs in one cohort. We may have used the term "study" ambiguously to refer both to the GLGC as a (meta-)study and to individual studies (i.e. included cohorts) that may be better referred to as samples, the term used by Taylor et al. The criticism in Taylor et al.'s elaborate simulation paper should not apply to our study as far as we understand, as our study is based on analysis results that combine over 30 independent samples in the GLGC, and not on one sample only.

Our study follows the recommendations for two-sample MR design and interpretation outlined by Haycock, Burgess and George Davey Smith (*Am J Clin Nutr.* 2016. PMID 26961927). This reference includes a discussion of Winner's curse (Beavis effect), which describe the bias from using overlapping genetic discovery and MR samples:

"[...] When GWASs report evidence of association for a trait at a specific genomic region, involving multiple, sometimes hundreds, of SNPs, they typically select the SNP with the smallest P value as the lead SNP and do not report the associations for other significant SNPs. This practice generally leads to overestimation of the SNP-trait effect, also known as the winner's curse or Beavis effect (34). The overestimation occurs because of chance correlation between SNPs and confounders in the GWAS discovery stage. If the GWAS discovery and MR studies are independent, the winner's curse will not affect the power or size of a causal hypothesis test, but it will bias MR estimates toward the null. To illustrate this, consider the Wald ratio, a common approach for deriving causal estimates from summary data with a single SNP. The Wald ratio is the coefficient of the SNP-outcome association divided by the coefficient of the SNP-exposure association. Thus, overestimation in the denominator, e.g., due to winner's curse, will result in an underestimation of the ratio but only when both samples are independent. When the discovery and MR analysis samples are the same, both the numerator and denominator will be overestimated."

In our two-sample MR design with separate samples (or be with 1-2% overlap) for the SNP-lipid and SNP-breast cancer associations, the bias from identical genetic discovery and SNP-exposure samples might bias results **towards the null** (i.e. for individual variants increase the denominator in the instrumental variable analysis estimate). Any effect detected in our study should therefore be less likely be a false positive finding, although we may be biased towards missing true positive signals and underestimating the strength of detected effects.

Taylor et al. (PMID 24388127) also show that SNPs should be selected based on genome-wide significant associations, and not through either a "candidate gene" approach or by employing arguments about biological plausibility of genetic variants' biological functions. We fully concur with this advice in our study by employing genome-wide significance thresholds in multi-sample summary results in order select SNPs.

Taylor et al. moreover caution that the measured exposure in the MR study should reflect the actual phenotype of interest as closely as possible (they use the example of whether reported-cigarettes-per-day is a good measure for lifetime smoking exposure). This type of measurement bias should not apply to our study that used objectively measured fasting plasma lipids levels and physician-diagnosed breast cancer episodes as measured variables.

The ideal two-sample MR set-up would involve three wholly independent samples for (1) genetic discovery, (2) the SNP-exposure arm in MR (the denominator in the classic Wald ratio estimate for single variants), and (3) the SNP-outcome arm (the numerator). The current convention in summary-level two-sample MR in the absence of enough separate large datasets is to use the same consortium data for genetic discovery and the SNP-exposure arm in MR (examples using the GLGC dataset: Harrison, Holmes, Burgess et al. *JAMA Cardiol.* 2018. PMID 29188294; Burgess & Davey Smith *Ophthalmology* 2017. PMID 28456421). It is crucial to avoid sample overlap between the SNP-exposure and SNP-outcome samples in MR (e.g. Burgess et al. *Genet Epidemiol.* 2016. PMID 27625185, and Taylor et al. *Econ Hum Biol.* 2014. PMID 24388127). We have now moved the section on

sample overlap from S1 Text to the Methods section and added further details. The overlap between both samples in our study is negligible:

Methods, page 9:

Sample overlap

Participant overlap between the samples used to estimate genetic associations with the exposure and the outcome in two-sample MR can bias results³⁷. A careful comparison of the samples included BCAC and GLGC showed one common cohort (EPIC), which accounted for 2.9% of cases and 3.3% of control persons in BCAC, and for 1.7% (1.0% if only considering women) of participants in GLGC. Based on a simulation study on the association between sample overlap and bias in instrumental variable analysis³⁷, this degree of overlap (< 5%) is unlikely to influence results.

In summary, the Reviewer concerns are highly justified. The negligible sample overlap should not have affected the results. Overlapping genetic discovery and MR SNP-exposure samples may have biased our study toward missing true positive signals - any significant discovered effects, however, should be unaffected or may even be underestimated.

The authors rightly acknowledge the ongoing controversy about the role of HDL and lipids levels in general to the risk of developing breast cancer. A stronger point in discussion explaining discrepancies and how this research might push the field in the right direction would strengthen the article.

Author response:

We thank the Reviewer for asking for a more thorough discussion of the discrepant results between previous (observational) studies on HDL and the contribution of our genetic evidence. In light of the revised results, we now discuss the robust harmful effects of LDL on page 15 in the Discussion more thoroughly than previously. As suggested by Reviewer, we have also added a more detailed discussion on the harmful effect of HDL to embed the contribution of our genetic evidence into the debate on the role of HDL in breast cancer.

Discussion, pages 15-16:

Genetically raised HDL-cholesterol was associated with increased risk of ER-positive breast cancer. **Observational studies have reported inconsistent results on associations between HDL and breast cancer, including null effects^{5, 7, 39, 40}, inverse associations in post-menopausal women⁶, and unidirectional associations in pre-menopausal women⁴¹. The effect of HDL-cholesterol on breast cancer risk in our study concurs with a non-significant association in an earlier genetic study²⁴, and the absence of a stronger effect in our summary-level data may relate to lack of**

power and an inability to stratify participants by menopausal status. Another explanation could be different effects depending on the metabolic health of the individual, as *in vivo* studies found evidence that oxidation status of HDL-cholesterol in a normo- or hyperlipidaemic contexts may determine its effects on promoting breast cancer metastasis⁴².

Taken together, laboratory studies have demonstrated that lipoprotein fractions affect breast cancer growth both directly and as precursors for cholesterol metabolites^{42, 43}, and observational studies in women have hitherto not been able to consistently define potential effects. Our study provides genetic evidence of a harmful association between raised LDL-cholesterol and breast cancer occurrence, as well as a suggestive harmful effect of raised HDL-cholesterol.

Minor:

please use estrogen or oestrogen consistently throughout the paper.

Author response:

Thank you, we use "estrogen" to chime with the usual abbreviation "ER" throughout the manuscript now and have changed inconsistent spellings in the earlier version. We use British English spelling throughout the manuscript, which requires "oestrogen" of course. We decided to use the exceptional American English spelling of "estrogen" since stating "oestrogen receptor (ER)" or alternatively using "OR" as abbreviation felt unsatisfactory to us. We are more than happy to change this in line with editorial and reviewer advice.

Reviewer #2:

Nowak et al. perform two-sample summary-based multi-instrument MR on blood lipids and breast cancer risk using the largest GWAS summary statistic datasets to date for lipids (GLGC) and breast cancer. They show that genetically-raised HDL-C was associated with higher risk of breast cancer and estrogen receptor-positive breast cancer, with a specific genic effect on CETP variants. Genetically lowering LDL cholesterol in statin target genes, PCSK9 inhibitors and ezetimide may have protective effects. The study is a well executed study that leverages the largest GWAS summary datasets to date for these traits. This work is relevant for breast cancer and has important implications for ongoing clinical trials. Below are my comments:

Strengths

1. leverages appropriate MR methods and approaches to infer causality of blood lipids on breast cancer
2. leverages largest GWAS summary datasets for both these traits
3. Well-written and concise

Limitations

4. The strength of statistical evidence of the causal relationship of HDL-C and breast cancer, and that of LDL-C and breast cancer, is consistent but the statistical evidence is moderate. One would have liked to see stronger evidence in the MR tests to more confidently say whether such causal relationships truly exist (re. more replication but its understood that this is sometimes difficult to find more datasets).

Author response:

We thank the Reviewer for the concise and highly constructive criticisms. We fully agree that ideally our results should be replicated in an entirely independent dataset. In our study, we make use of the largest available genetics datasets for both breast cancer risk (BCAC, published in late 2017) and blood lipid levels (GLGC). Unfortunately, we are unaware of accessible samples with sufficient number of participants with breast cancer (e.g. > 10,000) that are not already included in either of the consortia to replicate our results. This is a common problem of current MR studies that require large sample sizes, and will likely improve over the next ten years with the growing availability of genetic datasets of sufficient size (e.g. with accumulating case numbers in the UK Biobank). We have now revised the analysis in line with the Reviewer's suggestions as detailed below. We are more confident in the results of harmful effects of raised LDL and HDL now, as the revised analyses more strictly exclude pleiotropy and heterogeneity. The P-values for some of the estimates may not reach very low levels. However, we believe that an over-reliance on traditional null hypothesis significance testing may underappreciate potentially important effects that we consider worth bringing to the field's attention, particularly as our results on LDL and HDL

clarify the effect directions that observational studies have been inconsistent about. We have been careful to caution against over-interpretation of our study's results. For example:

Introduction, page 5:

Mendelian randomization makes several assumptions that are often difficult to ascertain, including the absence of genetic effects on the outcome that are independent of the exposure (absent horizontal pleiotropy)²⁰. It can thus only provide preliminary evidence of causality that may inform subsequent intervention studies, drug monitoring and public health approaches²¹.

Discussion, page 15:

Instrumental variable analysis with HDL-cholesterol-raising variants in *CETP* suggested a small risk-increasing effect on breast cancer **and ER-positive disease**; however, possible bias from pleiotropy cannot be excluded.

Discussion, page 17:

Whether pharmacological *CETP* inhibition could affect breast cancer risk remains uncertain, as lifelong genetic **effects** and **consequences of pharmacological intervention in mid-life** may differ²¹.

Discussion, page 19:

An early clinical trial of ezetimibe **raised** concerns that combination therapy with statins might be associated with increased risk of cancer. Subsequent longer follow-up and comparisons across **other** clinical trials, however, found no association with raised cancer risk^{48,49}. Our findings **indicate of a possibly protective effect agree with an earlier smaller genetic study¹⁶, but inconsistent estimates between Egger and inverse variance-weighted MR in our analysis indicates violations of model assumptions that do not allow for conclusive interpretations.**

Discussion, page 20:

Limitations include our inability to replicate results in independent datasets, concerns about pleiotropy from (un)measured confounders, possible weak instrument bias and lack of power for drug analyses. **A bias toward the null because of Winner's curse⁵⁰ as genetic discovery had been implemented in the same dataset used to estimate exposure associations cannot be excluded.** MR assesses the life-long effects of genetic variation and cannot be directly compared to pharmacological inhibition in adult life. Analysis accounted for population stratification (principle components and restriction to European ethnicity) and pleiotropy (MR Egger), but remaining sources of bias such as canalization cannot be ruled out. Finally, we could not assess the influence of menopausal status and our results only apply to women of European ethnicity.

5. The restrictive MR approach excludes pleiotropy with other lipid traits using genome-wide significant cutoff. One could use a more relaxed P-value such as $P < 0.05$ to more strictly exclude pleiotropic effects on these traits.

Author response:

We thank the Reviewer for the suggestion. In our revised analysis, we have now excluded lipid variants with associations at $P < 0.001$ with any of the other two lipids (in the original analysis, we excluded at $P < 5e-08$). As pointed out by the Reviewer, this provides a stricter exclusion of pleiotropic effects from other lipid traits. We did not exclude at the suggested level of $P < 0.05$ for two reasons. First, we felt that a very strict pleiotropy threshold might limit the biological plausibility of the potential causal effects of lipid fractions somewhat, given that all three fractions have substantial genetic, biological and observational correlations. We would expect that genetic variants that robustly affect one lipid fraction might also share some less pronounced associations with other lipid fractions, and we consider a less stringent threshold of $P < 0.001$ a reasonable compromise. Secondly, excluding pleiotropic associations at $P < 0.05$ left fewer than 10 variants for triglyceride levels. Whilst not a problem for MR in general, most genetic variants in our study have small effect sizes and individually explain little variance. Combining too few small-effect instruments in methods like MR Egger might lead to high imprecision and susceptibility to bias and noise. Altogether, we consider the pleiotropy exclusion threshold of $P < 0.001$ a sound compromise in order to assess individual lipid fractions' effects. We have changed major passages in the Methods, Results, and Discussion sections for reflect the revised analyses and results, for example:

Methods, page 6:

Blood lipids. We extracted association statistics for LDL-cholesterol, HDL-cholesterol and triglycerides in GLGC for 185 genetic variants in 157 loci previously demonstrated to be associated with at least one lipid fraction²⁷. We constructed two genetic instruments for each lipid. First, we selected all genome-wide significant ($P < 5 \times 10^{-8}$) variants associated with each lipid for *comprehensive* MR (76 variants for LDL-cholesterol, 85 for HDL-cholesterol, 51 for triglycerides). Second, **to reduce possible pleiotropic effects we excluded** in each selection those variants that were associated at $P < 0.001$ with any of the other two lipids for *restrictive* MR (**44** variants for LDL-cholesterol, **28** for HDL-cholesterol, **4** for triglycerides).

Fig. 2 caption:

Figure 2. Causal estimates of blood lipid levels on risk of all, ER-positive and ER-negative breast cancer. Inverse variance-weighted instrumental variable analysis using genome-wide significantly associated independent variants as instrumental variables for each lipid. Results following exclusion of variants associated at $P < 0.001$ with any of the other lipids and following removal of outlier variants ($P < 0.05$ in MR-PRESSO) are shown. Causal estimates express the

change in odds ratio (OR) per standard deviation (SD) increment in lipid concentration.

6. proxies for drug targets - rather than using variants with LD $r^2 < 0.40$ to address correlated IVs, one can consider using explicit approaches that takes into account the correlation between IVs (see PMID: 26661904)

Author response:

We thank the Reviewer for this suggestion and have revised the analysis. We now select genome-wide lipid-associated variants in the drug target genes and iteratively retain the top SNPs in LD $r^2 < 0.4$ with all other selected variants (we had previously used different r^2 thresholds for different genes following previous publications but without clear justification). We also now implement the IVW and Egger methods with explicit modelling of the correlation between the variants using the method suggested in the reference quoted by the Reviewer (Burgess et al. *Stat Med.* 2016, PMID 26661904). We have added this study as Reference 32 and explain the revised analysis as follows in the manuscript (of note: In the earlier version we used the Broads' SNAP tool to obtain the LD matrix. SNAP has in the meantime been taken down because of increasingly dated datasets and we now use the NIH/NCI's LDlink, which provided slightly different r^2 estimates than before):

Methods, pages 6-7:

Proxies for drug targets. To assess potential causal effects of changes in lipid levels due to pharmacological intervention, we selected polymorphisms within ± 100 base pairs of genes encoding drug targets that were genome-wide significantly associated with the target lipid and in low linkage disequilibrium with each other. Variants were ranked by P -value for lipid association in GLGC and iteratively selected in the order of increasing P -value provided they were in low linkage disequilibrium ($r^2 < 0.4$) with variants selected in preceding steps. We obtained pairwise linkage disequilibrium based on Phase 3 (Version 5) of the 1000 Genomes Project combined European reference sample via LDlink²⁸.

Methods, page 8:

For drug target MR with variants in moderate linkage disequilibrium ($r^2 < 0.4$), we implemented the inverse variance-weighted and Egger methods with explicit modelling of correlations between genetic variants according the method suggested by³² as implemented in the *MendelianRandomization* software³³.

Results, page 13:

We implemented inverse variance-weighted MR and MR Egger method with consideration of the correlation between genetic variants using seven LDL-

cholesterol-associated variants in *PCSK9*, three LDL-associated variants each for *NPC1L1* and *HMGCR*, and six variants for *LDLR* (S2 Table).

7. Cochran's Q test was used to test for heterogeneity between IVs. The original method has been shown to result in type 1 error rate (PMID: 29686387 and biorxiv: <https://www.biorxiv.org/content/early/2018/02/27/159442>). Please ensure that the modified version of the Q test (Q' test) is being used, which corrects for this inflated error rate.

Author response:

We thank the Reviewer for their valuable suggestion and have now replaced Cochran's Q test with the Q' test suggested by Bowden et al. (*biorxiv* 2018. <https://www.biorxiv.org/content/early/2018/02/27/159442>) and the MR-PRESSO test by Verbanck et al. (*Nat Genet.* 2018. PMID 29686387). We have also added both references to the manuscript as Ref. 35 and 36, respectively.

Methods, page 8:

To assess heterogeneity between individual genetic variants' estimates, we used the Egger intercept test³⁴, the Q' heterogeneity statistic³⁵, and the MR pleiotropy residual sum and outlier (MR-PRESSO)³⁶ test. The Q' statistic uses modified 2nd order weights that are a derivation of a Taylor series expansion and take into account uncertainty in both numerator and denominator of the instrumental variable ratio (this eases the no-measurement-error, NOME, assumption)³⁵.

Discussion, page 11:

Following exclusion of pleiotropic variants in restrictive MR (S4 Table), raised LDL-cholesterol was associated with higher risk of any breast cancer (OR 1.12, 95% CI, 1.02-1.23, $P = 0.017$) and ER-positive breast cancer (OR 1.17, 95% CI, 1.05-1.29, $P = 0.004$) with consistent estimates across the Egger and median methods but evidence of remaining heterogeneity (Q' P -values < 10⁻⁴). Raised HDL-cholesterol had no clear association with breast cancer risk (OR 1.08, 95% CI, 0.96-1.21, $P = 0.198$) with significant remaining heterogeneity (Q' P -value = 0.003). There was evidence of an effect of raised HDL-cholesterol on increased ER-positive breast cancer risk (OR 1.13, 95% CI, 1.01-1.26, $P = 0.028$, Q' P -value = 0.169).

8. Cooks distance was used to identify outliers; however, a recent study (PMID: 29686387) showed that Cooks distance performs less than ideal in detecting correct outliers; instead, one can use the Q' outlier test

(<https://www.biorxiv.org/content/early/2018/02/27/159442>) or MR-PRESSO (PMID: 29686387)

After removal of outliers, one can again perform test of heterogeneity between IVs to ensure that heterogeneity is no longer significant after removal of outliers in the outlier-corrected MR test. This can provide reassurance that heterogeneity is not an issue when assessing the causal estimate in the outlier-corrected MR test.

Author response:

We agree and thank the Reviewer for their suggestions. To assess heterogeneity, we had relied on the Egger intercept and Cochran's Q tests that may be biased by violations of the untestable InSIDE condition and inflated type 1 error, respectively (as demonstrated in both references highlighted by the Reviewer). The study by Verbanck et al. (*Nat Genet.* 2018. PMID 29686387) elegantly demonstrates the shortcomings of heterogeneity/outlier testing with Cook's distance and Cochran's Q. We have now revised the analysis and use both the Q' test and the MR-PRESSO method as global heterogeneity tests. We use the MR-PRESSO outlier test to identify outlying variants in cases of significant heterogeneity and we have also added the outlier-corrected retest for heterogeneity as suggested by the Reviewer. We summarize the study workflow in the new Fig. 1 and have revised major parts of the manuscript to reflect these changes, for examples:

Methods, page 8:

To assess heterogeneity between individual genetic variants' estimates, we used the Egger intercept test³⁴, the Q' heterogeneity statistic³⁵, and the MR pleiotropy residual sum and outlier (MR-PRESSO)³⁶ test. The Q' statistic uses modified 2nd order weights that are a derivation of a Taylor series expansion and take into account uncertainty in both numerator and denominator of the instrumental variable ratio (this eases the no-measurement-error, NOME, assumption)³⁵. The MR-PRESSO framework relies on the regression of variant-outcome associations on variant-exposure associations and implements a global heterogeneity test by comparing the observed distance (residual sums of squares) of all variants to the regression line with the distance expected under the null hypothesis of no pleiotropy³⁶. In case of evidence of horizontal pleiotropy, the test compares individual variants' expected and observed distributions to identify outlying variants. We used an implementation of MR-PRESSO (<https://github.com/rondolab/MR-PRESSO>) with default parameters to (i) test for global heterogeneity; (ii) if significant at $P < 0.05$ identify and remove outliers; and (iii) retest to evaluate if outlier removal had resolved heterogeneity. We consider as results causal estimates that agree in direction and magnitude across MR methods, pass nominal significance in inverse variance-weighted MR, and do not show evidence of bias from horizontal pleiotropy in heterogeneity tests. Power analysis is provided in **S1 Text**. Analyses were carried out with the *MendelianRandomization*³³, *TwoSampleMR* and MR-PRESSO³⁶ packages in R version 3.3.2 (2016-10-31).

Results, page 11-12:

We applied the MR-PRESSO method to the restrictive MR models to identify and remove outlying variants followed by retesting for heterogeneity (**Figure 2, S5 Table**). In inverse variance-weighted MR following the removal of outliers, raised LDL-cholesterol had a risk-increasing effect on breast cancer (OR 1.09, 95% CI, 1.02-1.18, $P = 0.020$, Q' P -value = 0.102) and ER-positive breast cancer (OR 1.14, 95% CI, 1.05-1.24, $P = 0.004$, Q' P -value = 0.124) and no association with ER-negative disease ($P = 0.577$). Raised HDL-cholesterol had no nominally significant association with either breast cancer risk (OR 1.07, 95% CI, 0.97-1.19, $P = 0.171$, Q' P -value = 0.090) or ER-negative disease (OR 1.09, 95% CI, 0.91-1.30, $P = 0.365$, Q' P -value = 0.108), but appeared associated with increased risk of ER-positive disease (OR 1.13, 95% CI, 1.01-1.26, $P = 0.037$, Q' P -value = 0.169). Genetically predicted triglyceride levels were not associated with any of the outcomes.

- 9. Can the authors please confirm if there is (and if so, approximately how much) participant overlap between the GLGC dataset and the Breast cancer dataset. Overlap of samples can potentially increase the false positive rate in two-sample MR (see PMID: 27625185).**

Author response:

We thank the Reviewer for allowing us to elaborate on sample overlap (which we had buried in the Supplementary Text before). It is crucial to avoid sample overlap between the SNP-exposure and SNP-outcome samples in MR (e.g. Burgess et al. *Genet Epidemiol.* 2016. PMID 27625185, and Taylor et al. *Econ Hum Biol.* 2014. PMID 24388127). The Reviewer rightly points out missing information on sample overlap in the main text. We have now moved the section on sample overlap from S1 Text to the Methods section and added further details. The overlap between both samples in our study is negligible and we do not think that it biased the results.

Methods, page 9:

Sample overlap

Participant overlap between the samples used to estimate genetic associations with the exposure and the outcome in two-sample MR can bias results³⁷. A careful comparison of the samples included BCAC and GLGC showed one common cohort (EPIC), which accounted for 2.9% of cases and 3.3% of control persons in BCAC, and for 1.7% (1.0% if only considering women) of participants in GLGC. Based on a simulation study on the association between sample overlap and bias in instrumental variable analysis³⁷, this degree of overlap (< 5%) is unlikely to influence results.

Minor comment

10. the reference for selecting 185 genetic variants in 157 loci should include PMID: 24097064

Author response:

We thank the Reviewer and apologise for the misattribution. We had now added the reference to Do et al. (*Nat Genet.* 2013. PMID 24097064) as reference 27 to manuscript.

Methods, page 6:

Blood lipids. We extracted association statistics for LDL-cholesterol, HDL-cholesterol and triglycerides in GLGC for 185 genetic variants in 157 loci previously demonstrated to be associated with at least one lipid fraction²⁷.

Reviewers' Comments:

Reviewer #1:

Remarks to the Author:

I appreciate the thorough discussion and answers to questions which arose during my review. I have no further suggestions at this point.

Reviewer #2:

Remarks to the Author:

The authors have done have a nice job of adequately addressing most of my comments. I only have two comments based on the revisions:

1) I think the MR-PRESSO method should be applied to the comprehensive MR analysis (all SNVs) which will allow the method to identify pleiotropic outliers on all SNVs (in addition to in the setting of restrictive MR where certain SNVs are removed due to a P-value cutoff with other lipid traits).

2) line 227: typo: outlying variants should be outlier variants.

Reviewer #1:

I appreciate the thorough discussion and answers to questions which arose during my review. I have no further suggestions at this point.

Author response:

We thank the Reviewer for their contributions to improving our study.

Reviewer #2:

The authors have done have a nice job of adequately addressing most of my comments. I only have two comments based on the revisions:

1) I think the MR-PRESSO method should be applied to the comprehensive MR analysis (all SNVs) which will allow the method to identify pleiotropic outliers on all SNVs (in addition to in the setting of restrictive MR where certain SNVs are removed due to a P-value cutoff with other lipid traits).

Author response:

We thank the Reviewer for their help in improving our study. In line we their suggestion, we have now applied MR-PRESSO to the all-SNV model as well (in addition to the model excluding pleiotropic SNVs associated with other lipids at $p < 0.001$). We now include these results as Supplementary Table 6. The causal estimates from this analysis agree in direction and magnitude with the previous results - but in all cases, removal of outlier variants leaves significant unexplained heterogeneity (evaluated by both, MR-PRESSO global test and the Q^* test). Since the more restrictive approach of additionally excluding variants based on the $P < .001$ pleiotropy criterion resolved the heterogeneity after outlier removal, we consider mostly those findings in the Discussion and mention the new results as supportive but somewhat uncertain estimates of lipid effects in the Results, page 8:

Application of MR-PRESSO to the comprehensive selection without exclusion of variants associated with other lipids produced causal estimates in the same direction, but there was significant ($P < 0.05$) remaining heterogeneity after exclusion of outliers in all cases (**S6 Table**).

2) line 227: typo: outlying variants should be outlier variants.

Author response:

Thank you for spotting our mistake. We have corrected it.

Reviewers' Comments:

Reviewer #2:

Remarks to the Author:

The authors have addressed my comments. I have no further comments.

Reviewer #2:

The authors have addressed my comments. I have no further comments.

Author response:

We thank both Reviewer #2 and Reviewer #1 for their constructive comments that have significantly improved our manuscript.